# The Effect of Obesity on Fractional Exhaled Nitric Oxide in School-Aged Children

**DOI:** 10.3390/children9091406

**Published:** 2022-09-16

**Authors:** Kamil Barański, Krzysztof Kocot

**Affiliations:** 1Department of Epidemiology, Faculty of Medical Sciences in Katowice, Medical University of Silesia in Katowice, 40-055 Katowice, Poland; 2Department and Clinic of Pediatric Cardiology, Faculty of Medical Sciences in Katowice, Medical University of Silesia in Katowice, 40-752 Katowice, Poland

**Keywords:** fractional exhaled nitric oxide, body mass index, children, inflammation, anthropometric measurement

## Abstract

Background: Fractional exhaled nitric oxide (FeNO) is recognized as a biomarker of eosinophilic inflammation. Current literature shows evidence that FeNO is influenced by many factors. Obesity is a chronic inflammatory state. In this study, we considered obesity as a potential factor that influences FeNO levels. The aim of the study was to analyze the association between body mass index (BMI, body mass (kg)/height (m)^2^) and FeNO levels in a young group of children. Methods: The participants in the study were 506 school-aged children who were randomly selected from primary schools located in Silesian Voivodship (Poland). The modified version of the Study of Asthma and Allergies in Childhood (ISAAC) questionnaire was used to assess the respiratory system status of children. FeNO was measured in 447 children according to European Respiratory Society and America Thoracic Society (ERS/ATS) recommendations. Body mass and height were measured by a medical body composition analyzer. BMI was defined and interpreted with Palczewska’s percentile charts. Results: In the study group there were 49.9% (*n* = 223) boys and 50.1% (*n* = 224) girls. The frequency of normal BMI was 76.8% (*n* = 172), overweight 13.7% (*n* = 31) and obesity 9.4% (*n* = 21) in girls, while the normal BMI was found in 71.3% (*n* = 159), overweight 11.6% (*n* = 26) and obesity 17% (*n* = 38) in boys, the differences not statistically significant (*p* = 0.05). The mean FeNO value in children with obesity was 16.1 ± 12.5 ppb, in children with normal BMI 15.8 ± 15.5 ppb and the lowest FeNO values were in children with overweight 15.3 ± 13.0 ppb; *p* = 0.9. The FeNO values after adjusting for age, sex, BMI and symptoms from respiratory system were depended on age and respiratory symptoms only. Conclusions: In 6–9 year old school children, FeNO levels are associated with age and health in relation to the respiratory system. The BMI should not be included when considering reference values for FeNO.

## 1. Introduction

Fractional Exhaled Nitric Oxide (FeNO) has been introduced as a biomarker of eosinophilic airway inflammation. Because of this, it is widely used for the diagnosis and monitoring of asthma [1]. Moreover, it has been proved that FeNO increases when an inflammatory process occurs in various organs [2].

It is also known that FeNO levels are dependent on factors such as sex [3], the status of asthma, allergic rhinitis, atopic dermatitis or allergy [4], age [5], lung function, physical activity [6], infection in last two weeks [7] and exposure to tobacco smoke [8]. In some studies, the association between body mass index and FeNO levels in children has also been introduced [9,10,11].

The purpose of the study came from the results of other studies. Current published studies have shown contradictory results regarding FeNO levels and obesity in children. A study conducted on the Turkish population suggests a positive correlation between BMI (body mass index–body mass (kg)/height (m)^2^) and FeNO level [10], while a study based on early US adolescents suggests that increased and decreased BMI is associated with lower FeNO levels [9]. The results from a study in school-aged Taiwanese children show that excess weight inversely affects FeNO in atopic but not in non-atopic children [11]. Regarding these facts, we decided to analyze the effect of BMI on FeNO level in a population of Polish children aged between 6–9 years.

## 2. Materials and Methods

### 2.1. Participants

The study was conducted between 2017–2020. The participants in the study were school-aged (6–9 years) children who were randomly selected from primary schools in 4 cities (Bytom, Chorzów, Tychy and Zabrze) located in Silesian Voivodship (Poland). All children whose parents or legal guardians signed informed written consent were included in the study and additional inclusion criteria were willingness (child) to participate in the study, fulfilling the questionnaire and presence at school on the day of the measurements.

### 2.2. Research Tools

The modified version of the Study of Asthma and Allergies in Childhood (ISAAC) questionnaire [12] was used to assess the respiratory system status of children. The following respiratory/allergic outcomes were included in the analysis: asthma, pneumonia, allergy, rhinitis and atopic dermatitis diagnosed by physician in the past. All diseases and symptoms were identified based on the parental answers. Asthmatic tendency was defined as attacks of dyspnea and chest wheezes not related to infectious disease (such as cold) during the last 12 months.

FeNO measurement (NIOX MINO device, Circassia, Stockholm, Sweden) was performed according to ERS/ATS recommendations [13]. The tests were applied by a single, trained and certified researcher. All children and parents were informed to avoid drinking and eating 1 h prior to FeNO measurement. Results of FeNO measurement were expressed in ppb. The measurement of FeNO was conducted from the second day of the week to decrease the risk of the impact of exposure to tobacco smoke (positive smoking status of parents).

Body mass and height were determined using the body composition analyzer (type IOI-353, Yawon).

### 2.3. Definitions

Overweight and obesity were defined according to percentile charts developed by Palczewska for the Polish population of children and adolescents [14]. Overweight was noticed when children’s BMI (after adjusting for sex and age) was between 90–97 percentile, while obesity was defined as BMI above 97 percentile. The description of the results considered two strategies: first, children divided into three BMI categories (normal weight vs. overweight vs. obesity); second, children divided into two categories (normal weight vs. overweight + obesity combined).

In the study group (*n* = 506), technically acceptable (FeNO measurements) were obtained in 447 children (89% success rate).

The study followed the rules described in the Declaration of Helsinki. Moreover, the protocol was approved by the Ethics Committee of the Medical University of Silesia in Katowice (decision no KNW/0022/KB1/37/IV/14/16/17/).

### 2.4. Data Analysis

The quantitative variables were presented as the arithmetical mean and standard deviation (±). The qualitative variables were presented as frequency (*n*) and percentage (%). The normal distribution was analysed by Shapiro-Wilk test. The statistical significance of differences between quantitative variables were assessed using the Wilcoxon test or Student test when appropriate. The relationship between qualitative variables was assessed by the Chi-square test or Fisher test. The analysis of associations between quantitative variables was assessed using the Spearman correlation. The associations between FeNO levels as a dependent variable with independent variables were assessed with multiple linear regression and two-way factorial ANOVA. The group of analyzed children was a representative sample of children from Silesian voivodship; however, considering the sample size in relation to overweight or obesity, these groups were not representative.

The *p* < 0.05 was considered statistically significant. All analyses were performed using the SAS statistical package (SAS Institute Inc., Cary, NC, USA, version 9.4).

## 3. Results

In the study group there were 49.9% (*n* = 223) boys and 50.1% (*n* = 224) girls. Both groups did not differ according to age: 7.4 ± 0.7 (arithmetic mean ± standard deviation) and 7.4 ± 0.8, respectively. The frequency of normal BMI (NB) in girls was 76.8% (*n* = 172), overweight (OV) 13.8% (*n* = 31) and obesity (OB) 9.4% (*n* = 21), while the normal BMI was found in 71.3% (*n* = 159) of boys, overweight in 11.7% (*n* = 26) and obesity in 17% (*n* = 38), the differences not statistically significant (*p* = 0.05). The results of anthropometric variables in children included in the study are shown in Table 1.

All children were assessed regarding their respiratory system status. There were no statistically significant differences between groups of normal BMI, overweight and obesity, in terms of ever diagnosed pneumonia, rhinitis, atopic dermatitis or any allergy.

In the whole group of children, there were 77 cases of ever-diagnosed pneumonia (the percentages are calculated in relation to specific BMI category; NB = 317, OV = 50, OB = 54) *n* = 57 (18%) NB, 9 (18%) OV, 11 (20%) OB; *p* = 0.9), 106 cases of rhinitis (79 (25%) NB, 13 (24%) OV, 14 (25%) OB; *p* = 0.9), 86 cases of atopic dermatitis (*n* = 62 (19.5%) NB, 12 (22.2%) OV, 12 (21.4%) OB; *p* = 0.8), while 127 children had any allergy diagnosed by a physician (*n* = 88 (28%) NB, 17 (30.4%) OV, 22 (39.2%) OB; *p* = 0.2).

Considering the respiratory status of children, in the whole group there were 349 (78%) who did not have asthmatic tendency, defined as presence of symptoms of wheezing and dyspnea currently or in the last 6 months (*n* = 265 (80%) NB, 44 (77.2%) OV, 40 (67.8%) OB. Asthma was present statistically significantly more often in children with obesity than in those with normal BMI or in overweight (*n* = 30 (9.1%) NB, 1 (1.8%) OV, 6 (10.2%) OB; *p* = 0.02).

In terms of respiratory symptoms (current or in past six months), there were no statistically significant differences between groups of normal BMI, overweight and obese. However, when analyzing the groups of normal BMI vs. overweight and obese combined, the latter group statistically significantly more often presented wheezes and exercise induced dyspnea (Table 2).

Wheeze was reported in 74 children (*n* = 47 (14.6%) NB, 12 (21.8%) and 15 (26.3%) OB; *p* = 0.05), dyspnea during the day in 13 (*n* = 10 (3.2%) NB and 3 (5.5%) OV; *p* = 0.2), night dyspnea in 17 (*n* = 12 (3.6%) NB, 3 (5.6%) (*n* = 3) OV, 2 (3,5%) OB; *p* = 0.8) and dyspnea induced by exercise in 32 (*n* = 17 (5.5%) NB, 5 (9.2%) OV, 10 (17.8%) OB, *p* = 0.005)

The symptoms of rhinitis were reported in 111 children (*n* = 76 (22.8%) NB, 15 (30.6%) OV, 20 (37%) OB; *p* = 0.1), while atopic dermatitis was present in 69 (*n* = 47 (14.6%) NB, 13 (23.6%) OV, 13 (15.8%) OB (*n* = 13); *p* = 0.2).

The frequency of respiratory symptoms and diagnoses in children with normal body weight vs. overweight and obesity combined are presented in Table 2.

### FeNO Levels

Boys had slightly higher FeNO values than girls, 16.0 ± 12.1 ppb vs. 15.6 ± 17.1 ppb, but the differences were not statistically significant (*p* = 0.06). When considering BMI categories, the highest FeNO values were found in children with obesity, 16.1 ± 12.5 ppb, rather than in children with normal BMI, 15.8 ± 15.5 ppb and the lowest FeNO values were in children with overweight 15.3 ± 13.0 ppb; *p* = 0.9. The FeNO values in the group of children with normal BMI and combined categories of overweight and obesity in relation to ever diagnosis and current symptoms from respiratory system are presented in Table 3.

The result of FeNO values in children with ever-diagnosed pneumonia and with NB were 18.3 ± 17.9 ppb, in children with OV 20.2 ± 26.3 ppb and in OB 18.9 ± 18.8 ppb. In children with atopic rhinitis and NB: 20.0 ± 23.2 ppb, OV: 13.7 ± 11.5 ppb, OB: 24.5 ± 20.3 ppb, atopic dermatitis and NB: 21.4 ± 26.3 ppb, OV: 15.0 ± 11.0 ppb, OB: 17.5 ± 10.6 ppb, any allergy and NB: 18.1 ± 22.5 ppb, OV: 20.4 ± 20.8 ppb, OB: 17.8 ± 10.1 ppb. In those with no symptoms from respiratory system and NB: 13.8 ± 9.0 ppb, OV: 16.7 ± 14.4 ppb, OB: 13.0 ± 8.1 ppb, asthmatic tendency and NB: 22.1 ± 20.9 ppb, OV: 10.7 ± 3.7 ppb, OB: 19.0 ± 13.1 ppb, asthma and NB: 26.5 ± 35.5 ppb, OV: 11 ppb, OB: 24.2 ± 18.3 ppb.

FeNO values according to respiratory symptoms were: wheeze in the chest and NB: 26.6 ± 32.0 ppb, OV: 11.0 ± 3.6 ppb, OB: 24.2 ± 18.3 ppb, dyspnea during the day and NB: 16.9 ± 14.7 ppb, OV: 10 ± 2 ppb, dyspnea at night and NB: 19.2 ± 15.5 ppb, OV: 11.0 ± 1 ppb, OB: 32.0 ± 22.6 ppb, exercise-induced dyspnea and NB: 20.0 ± 15.7 ppb, OV: 10.8 ± 4.7 ppb, OB: 19.9 ±5.4 ppb, allergic rhinitis and NB: 23.6 ± 29.4 ppb, OV: 15.2 ± 10.0 ppb, OB: 14.6 ± 5.4 ppb, atopic dermatitis and NB: 19.2 ± 23.1 ppb, OV: 15.4 ± 11.6 ppb, OB: 16.6 ± 12.6 ppb. Only the differences between children with current symptoms of wheezing in the chest and according to level of BMI were close to statistically significant results (*p* = 0.06).

The FeNO values did not correlated with BMI but a weak correlation was found in relation to age: R: 0.13; *p* = 0.005. The same results were noticed in children without symptoms of the respiratory system R: 0.02; *p* = 0.6, R: 0.16; *p* = 0.001, for BMI and age respectively. In children with asthmatic tendency, the correlation of FeNO values with BMI was R: −0.03; *p* = 0.8 and for age R: 0.02; *p* = 0.8, while in children with asthma R: 0.01; *p* = 0.9 and R: 0.21; *p* = 0.2, respectively.

Results of the multiple linear regression indicated that there was a collective significant effect between the sex, age, BMI and symptoms from respiratory system, R2 = 0.07, F(4, 442) = 9.40, *p* < 0.0001. The individual predictors were examined further and indicated that age (t = 2.78, *p* = 0.005) and symptoms from respiratory system (t = 5.40, *p* < 0.0001) were significant predictors in the model.

## 4. Discussion

Our study aimed to investigate the association between FeNO levels and body mass index in conditions of epidemiological study.

Hyper-nutrition that leads to obesity and overweight is a growing problem, especially in high income countries, and it affects both adults and children. According to the WHO European Regional Obesity Report, almost one-third of school-aged children have overweight or obesity [15]. Obesity is a state related to an increased inflammatory response of the body [16]. Adipose tissue in obese individuals produces high amounts of pro-inflammatory cytokines, such as Interleukin-6 (IL-6) or Tumor Necrosis Factor-α (TNF-α), while the secretion of adiponectin decreases, which results in chronic inflammation [16]. Apart from increased inflammation, high BMI is also related to increased oxidative stress [17]. Both those pathways may further lead to development of several diseases, such as diabetes, atherosclerosis with all its consequences and cancer [16]. Moreover, obesity alters the function of airways’ epithelial cells, lung immune cells and fibroblasts, inducing airway inflammation and eventually remodeling, an important risk factor of asthma onset and greater disease severity [18].

Due to this role of inflammation in the pathological mechanisms of obesity, high BMI might also affect the levels of FeNO, which is a marker of airways’ eosinophilic inflammation [1,2]. Considering the frequency of high BMI and the role of FeNO in the diagnosis and monitoring of asthma, it is important to determine, whether body mass affects FeNO measurements in children.

The results of our study do not support the hypothesis that FeNO variability corresponds with BMI variability. Our study confirms that FeNO levels in children with asthma, asthmatic tendency or without symptoms from respiratory system are not different when controlling for BMI. In addition, this does not differ between different BMI categories when not considering symptoms from the respiratory system, as found in school-aged patients from Turkey [10]. However, the same research team reported that BMI is positively correlated with FeNO level in the non-asthmatic [10]. In contrast, a study on a group of asthmatic and healthy adults showed, that only in those with asthma was BMI associated with reduced exhaled NO [19]. There are also other studies that described the relation of FeNO to the state of obesity and other inflammatory respiratory diseases in patients, but the number of participants in those studies was relatively small and they were also conducted on adults [20,21]. Our study, which does not confirm the association between BMI and FeNO levels, was conducted on a large group of primary school children.

It seems that age is a crucial variable that plays a role in determining the level of FeNO (probably due to physiological changes), which was confirmed by a similar study conducted previously in an Italian population of 10–16 year old schoolchildren [22]. Zhang et al. compared FeNO levels between children aged 6–9 and 10–15 years old and concluded that FeNO increases within the age group [23]. Another researcher that confirmed the impact of age additionally indicated the impact of sex [24]. We did not observe the effect of sex on FeNO levels. This may be partly because of the fact that, in our study, we included only primary school children, while Yao, et al. studied a group of children aged 5 to 18 years old [24]. Younger children might not present sex-dependent changes, that become significant only after the maturation process in teenagers and adults.

### Strengths and Limitations of the Study

Our study included a large number of participants and all measurements were performed by a single certified researcher with certified and acceptable devices. However, like most studies, we could not avoid the typical limitations of epidemiological studies. The first was related to the assessment of respiratory status. Each diagnosis was based on self-reported symptoms/disease declared in the questionnaire by the children’s parent or legal guardian. We were not able to measure drug use in children probably treated for asthma or with current symptoms from respiratory system. Moreover, we did not measure airway inflammation by blood tests or by inducing sputum; however, gold standards are difficult to implement, especially in young groups under investigation, which also increases the costs of the study. In addition, we do not have data about bronchial and alveolar exhaled nitric oxide separately. However, alveolar measurement uses the same technic as bronchial but needs more trials on different levels of flow, which could be too difficult for a young group of participants. We would like to underline that we performed all measurements at schools, which allowed us to generate a representative group of participants from the Silesian voivodship.

## 5. Conclusions

In conclusion, we do not confirm the association between BMI and FeNO levels in early school-aged children, or when considering the co-variability of the status of the respiratory system (no symptoms and no diagnosis vs. asthmatic tendency vs. asthma ever diagnosed by a physician). In addition, according to our results, in early school-aged children FeNO levels are not sex-dependent. The only predictors of FeNO were age of children (in our study 6–9 years) and respiratory symptoms. Thus, only age and the status of the respiratory system should be considered while estimating reference FeNO values in early school-aged children. This has a practical implications for FeNO measurements and interpretation, considering the increasing frequency of high body mass in children.

## Figures and Tables

**Table 1 children-09-01406-t001:** Anthropometric variables of children included in the study.

Variable		All Children *n* = 447	Normal Weight *n* = 331	Overweight and Obese *n* = 116	*p*-Value
		7.4 ± 0.8	7.4 ± 0.8	7.5 ± 0.8	0.1
Sex	Boys Girls	223; 49.9% 224; 50.1%	159; 48.0% 172; 52.0%	64; 55.2% 52; 44.8%	0.1
Body mass (kg)		27.7 ± 6.4	24.9 ± 3.8	35.7 ± 5.7	0.0001
Height (cm)		128.3 ± 7.2	127.1 ± 7.0	131.9 ± 6.5	0.0001

**Table 2 children-09-01406-t002:** Symptoms from respiratory system and diagnoses confirmed by a physician according to BMI status of children included in the study.

Variable		All Children *n* = 447	Normal Weight *n* = 332	Overweight and Obese *n* = 115	*p*-Value
Diagnosis of: (ever)	Pneumonia	77; 17.2%	57; 17.1%	20; 17.3%	0.7
	Atopic Rhinitis	106; 23.7%	79; 23.7%	27; 23.4%	0.9
	Atopic dermatitis	86; 20.1%	62; 18.6%	24; 20.8%	0.6
	Allergy	127; 28.4%	88; 26.5%	39; 33.9%	0.1
Respiratory system status	No symptoms	349; 78.1%	265; 79.8%	84; 73%	0.01
	Asthmatic tendency	61; 13.6%	36; 10.9%	25; 21.7%	
	Asthma	37; 8.3%	30; 9.0%	7; 6.0%	
Current symptoms (or past 6 months)	Wheeze	74; 16.5%	48; 14.4%	27; 23.4%	0.02
	Dyspnea day	13; 2.9%	10; 3.0%	3; 2.6%	0.5
	Dyspnea night	17; 3.8%	12; 3.6%	5; 4.3%	0.7
	Exercise-induced dyspnea	32; 7.1%	17; 5.1%	15; 13.0%	0.005
	Rhinitis	111; 24.8%	76; 22.8%	35; 30.4%	0.06
	Atopic dermatitis	69; 15.4%	49; 14.7%	22; 19.1%	0.2

**Table 3 children-09-01406-t003:** FeNO levels according to diagnosis and symptoms from respiratory system in children.

Variable		Normal Weight	Overweight and Obese	*p*-Value
Diagnosis of: (ever)	Pneumonia	18.3 ± 3	19.5 ± 21.9	0.8
	N	57	20	
	Rhinitis	20.0 ± 23.2	19.3 ± 17.2	0.4
	N	79	27	
	Atopic dermatitis	21.4 ± 26.3	16.2 ± 10.6	0.9
	N	62	24	
	Allergy	18.1 ± 22.5	18.9 ± 15.5	0.1
	N	88	39	
Respiratory system status	No symptoms	13.8 ± 9.0	15.0 ± 11.9	0.7
	N	265	84	
	Asthmatic tendency	22.1 ± 20.9	15.0 ± 10.5	0.2
	N	36	25	
	Asthma	26.5 ± 35.2	27.4 ± 22.5	0.3
	N	30	7	
Current symptoms (or past 6 months)	Wheeze in the chest	26.6 ± 32.0	18.3 ± 15.1	0.4
	N	47	27	
	Dyspnea during the day	16.9 ± 14.7	10.0 ± 2.0	0.3
	N	10	3	
	Dyspnea during the night	19.2 ± 15.5	19.4 ± 16.1	0.9
	N	12	5	
	Exercise-induced dyspnea	20.0 ± 15.7	16.8 ± 12.1	0.5
	N	17	15	
	Rhinitis	23.6 ± 29.4	15.0 ± 8.3	0.4
	N	47	22	
	Atopic dermatitis	19.2 ± 23.1	16.0 ± 12.0	0.4
	N	76	35	

## Data Availability

The data are available on a reasonable request from the corresponding author.

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
