# Peer review of "The Effect of Obesity on Fractional Exhaled Nitric Oxide in School-Aged Children"

_children, 2022, doi:10.3390/children9091406_

Round 1

Reviewer 1 Report

To the authors:

1.     General comments:

The manuscript entitled “The effect of obesity on Fractional Exhaled Nitric Oxide in school-aged children” is an interesting study analyzing the FeNO in 447 Polish children of approx. 8 years old. This a novel application that has not been fully covered before, and the aim was to find out the relationship of FeNO with obesity especially, and other factors such as age, sex, and respiratory conditions. The results seem solid, however, there are major comments that I consider important to change/discuss in the manuscript:

2.     Specific comments for revision: b) major.

a)    Manuscript. It is confusing that throughout the manuscript sometimes it is talked about body mass, body mass index and BMI (which has not been defined at all), please unify the concepts.

b)    Lines 43-44. Please review the sentence, low and high BMI are associated with low or high FeNO?

c)     Lines 100. It is not clear which differences were not statistically significant?

d)    Lines 105-106. Define the groups before, in the first time the groups are stated in the text.

e)    Review the age of the children, in the introduction it is stated from 6-10 y.o. while in the table 1, it is stated 7.4 +/- 0.8 (please also add the meaning after the +/-.

f)      Abstract. Define BMI.

g)    Lines 108-110. It is not clear how the percentages were calculated, foe example, for children of ever-diagnosed pneumonia NB with 57 children represent an 18%, the same as OV with only 9 children.

h)    Is there any correlation, taking into account only obese children, instead of grouping overweigh and obese?

Author Response

Dear Reviewer 1,

Thank You for the time and effort that you put into the improvement of the manuscript.

We hope that all corrections that we made will be satisfactory to You and that You will find the article suitable for the publication in the journal.

  1. It is confusing that throughout the manuscript sometimes it is talked about body mass, body mass index and BMI (which has not been defined at all), please unify the concepts.

Response: We standardized the used terms where it was possible. The definition of BMI was added in the text (body mass (kg) / height (m)2). At the same time we would like to mention that overweight and obesity was defined according to BMI percentile charts developed by Palczewska. We explained that in the methodology. Kindly see line 69-76.

  1. Lines 43-44. Please review the sentence, low and high BMI are associated with low or high FeNO?

Response: We changed term low and high for increased and decreased. The cited publication describes that if someone has lowered BMI (below normal BMI) or increased BMI (over normal BMI) then FeNO is decreased.

  1. Lines 100. It is not clear which differences were not statistically significant?

Response: We paraphrased the sentence for: The p<0.05 was considered statistically significant.

  1. Lines 105-106. Define the groups before, in the first time the groups are stated in the text.

Response: Done, as recommended. Thank you.

  1. Review the age of the children, in the introduction it is stated from 6-10 y.o. while in the table 1, it is stated 7.4 +/- 0.8 (please also add the meaning after the +/-.

Response: Thank you for very meticulous review. We made correction and explained the meaning of +/-.

  1. Define BMI.

Response: We defined BMI in the abstract (body mass (kg) / height (m)2). BMI was interpreted in accordance with Palczewska’s percentile charts.

  1. Lines 108-110. It is not clear how the percentages were calculated, foe example, for children of ever-diagnosed pneumonia NB with 57 children represent an 18%, the same as OV with only 9 children.

Response: We agree that the description is awkward, the percentage is calculated from specific subgroup, for example 57 NB children is the 18% of children from total NB group (317). 57/317. We fixed the description. 

  1. Is there any correlation, taking into account only obese children, instead of grouping overweigh and obese?

Response: When considering correlations only in specific subgroups, the BMI is correlated positively with the age.

Reviewer 2 Report

The authors look at the relationship between FENO and obesity. The comments are as follows:

- Redo the abstract.

- Line 40 to 47 seems like discussion. redo intro.

- Redo materials and methods separating it by sections.

- Check the tables and put them horizontally.

In the table OV and OB go together but then separately in the results. line 158- 160 upload it and put it after 141.

Author Response

Dear Reviewer 2.

Thank You for the time and effort that you put into the improvement of the manuscript.

We hope that all corrections that we made will be satisfactory for You and that You will find the article suitable for the publication in the journal.

- Redo the abstract.

Response: We reedited abstract. We hope it is satisfactory.

- Line 40 to 47 seems like discussion. redo intro.

Response: We edited the part of introduction. However, lines 40-47 seem adequate for the introduction, as they are describing contradictory results of other studies, which is important for the purpose of the study.

- Redo materials and methods separating it by sections.

Response: Done as recommended.

- Check the tables and put them horizontally.

Response: It seems to us that the horizontal orientation would make the tables less clear, however we understand that the Reviewer may find it the opposite. Therefore, we decided to leave the decision on the orientation of the tables to the Editor.

In the table OV and OB go together but then separately in the results. line 158- 160 upload it and put it after 141.

Response: In material and methods we described two strategies of description for results (normal weight vs. overweight vs. obesity and normal weight vs. overweight and obesity). We did as recommended, however we didn’t want to repeat the results in tables and in the text.

Reviewer 3 Report

This is an interesting study about Feno in children, although it does not carry groundbreaking novelties.

There are some flaws which should be corrected:

- Please adjust Table 1 (sex-boys-girls).

- Similarly, in table 2 please rotate the variables column.

- Again, please do the same as above in table 3.

- "The measurement of FeNO was conducted on the second 67 day of the week to decrease the risk of the impact of exposure to tobacco smoke"... what does this mean? Were some kids smoking or else?? Please be more specific.

- Did authors estimate sample size? Please specify in data analysis section.

- English needs to be quite deeply re-edited.

- One more limitation of this study is due to the absence of data about separate bronchial and alveolar exhaled nitric oxide. Please comment on that in discussion section. 

Author Response

Dear Reviewer 3.

Thank You for the time and effort that you put into the improvement of the manuscript.

We hope that all corrections that we made will be satisfactory for You and that You will find the article suitable for the publication in the journal.  

There are some flaws which should be corrected:

- Please adjust Table 1 (sex-boys-girls).

Response: we corrected this issue, thank you.

- Similarly, in table 2 please rotate the variables column.

Response: Edited as recommended.

- Again, please do the same as above in table 3.

Response: Edited as recommended.

- "The measurement of FeNO was conducted on the second 67 day of the week to decrease the risk of the impact of exposure to tobacco smoke"... what does this mean? Were some kids smoking or else?? Please be more specific.

Response: The aim of measuring FeNO on the second day of the week was to reduce the impact of passive smoking, as some of the parents smoked. We edited the sentence as recommended, to make it clear.  

- Did authors estimate sample size? Please specify in data analysis section.

Response: We added a sentence regarding the sample size.

- English needs to be quite deeply re-edited.

Response: Thank you for your suggestion. The manuscript was reviewed by the native speaker in order to improve the language.

- One more limitation of this study is due to the absence of data about separate bronchial and alveolar exhaled nitric oxide. Please comment on that in discussion section. 

Response: We added a sentence regarding this issue in the limitations. Thank you.

Reviewer 4 Report

The manuscript entitled “the effect of obesity on fractional exhaled nitric oxide in school-age children” by Baranski is a very interesting manuscript to overcome the obesity challenges in future generations. It is really needed to understand and awareness of obesity-related health issues for the next generation. Hence, this manuscript is suitable for publication in children's journals. However, the following minor comments need to be addressed by the author

The author used (1), (2), (3), and so on in the abstract. Which are not required as the author clearly mentioned the background and method and results and so on

The indication of the corresponding author along with the author's name is missing. Both authors do not have * but at end of affiliations, the author mentioned only the email of the corresponding author. Author needs to add * to the respective author in authors sections

The keywords, should not be abbreviated so add full details of FeNO and BMI or used other keywords

In the introduction, the author cited several papers in the second paragraph. Recommended to reduce and replace with suitable literature which covers the complete sentence instead of just a word

In the table author “sex” word is used vortically and wrong direction as it is moving from top to bottom. An author can keep a simple horizontal way or change the bottom to top direction

As mentioned in the above comments, similar changes are also applicable for table 2 and table 3 as well

Conclusions need to be elaborated

Author Response

Dear Reviewer 4.

Thank You for the time and effort that you put into the improvement of the manuscript.

We hope that all corrections that we made will be satisfactory for You and that You will find the article suitable for the publication in the journal.

The author used (1), (2), (3), and so on in the abstract. Which are not required as the author clearly mentioned the background and method and results and so on

Response: We corrected this as recommended.

The indication of the corresponding author along with the author's name is missing. Both authors do not have * but at end of affiliations, the author mentioned only the email of the corresponding author. Author needs to add * to the respective author in authors sections

Response: We indicated the corresponding author (K.Barański) as recommended.

The keywords, should not be abbreviated so add full details of FeNO and BMI or used other keywords

Response: Thank you, we corrected the keywords.

In the introduction, the author cited several papers in the second paragraph. Recommended to reduce and replace with suitable literature which covers the complete sentence instead of just a word

Response: Each article in this section is mentioned in the proper place and indicates the specific study that showed the influence of different factors on FeNO levels. If these were to be cited at the end of the sentence, it would be hard for the reader to find the specific article for the specific factor (age, allergy etc.). Therefore, we ask the Reviewer to agree for the current form.  

In the table author “sex” word is used vortically and wrong direction as it is moving from top to bottom. An author can keep a simple horizontal way or change the bottom to top direction

Response: Thank you, we corrected this, so that the word is written bottom to top direction.

As mentioned in the above comments, similar changes are also applicable for table 2 and table 3 as well

Response: We corrected this as recommended.

Conclusions need to be elaborated.

Response: Thank you, we added some information to the conclusions, we hope that the current version is suitable.

Reviewer 5 Report

Reviewer

Initial comments

This paper is important, but the Abstract could be better written so that we can already understand the work.

Abstract:

Comment:

Line 10.... Please remove one of the two signs from :

Line 12….. BMI

Please spell out BMI

Lines 16-17…. ERS/ATS  recommendations

Please spell out ERS/ATS

Line 17… Body mass or BMI?

Line 13…. The participants of the study were 506 school-aged children

Line 16… FeNO was measured in 447 children according to ERS/ATS

Why only in 447 if the total is 506 children? I needed to explain.

Line 18…. Results: In the study group there were 49.9% (n=223) boys and

50.1% (n=224) girls.

 506 school-aged children ……………223 +224 = 447

There are 59 left to complete the total of 506... Why?

Lines 20-21..... while the normal body mass was found in 71.3% (n=159) boys,

overweight 11.6% (n=26) and obesity 20 17% (n=38)

159+26+38= 223…..boys

And the girls?

 Lines 22-23-24……The FeNO values after adjusting for age, sex, BMI, and symptoms from respiratory system were depended on age and symptoms from respiratory system only.

Please, in Materials and Methods, the age of the patients was not included....

1. Introduction  

 Comment:

 Line 38…..between body mass and FeNO levels

body mass or BMI ?

 Line 39…well. [9-11].

Please remove the point after well

2. Materials and Methods 

Comment:

Lines 50-51…..The participants of the study were school-aged children who were …

But, what is the school age?

Line 69…Body mass

Body mass or BMI ?

Lines 77-78…In the study group (n=506) the technically acceptable (FeNO measurements) were 77 obtained in 447 children (89% of success rate).

Please, how many boys and girls?

Data Analysis

Comment:

It is suitable

1. Results

Comment:

Line 95....

Please isn't it number 3?

FeNO levels

Comment:

It is suitable

4. Discussion

Comment:

It is suitable

5. Conclusions

  Comment:

It is suitable

References

Comment:

It is suitable

Thank you

Author Response

Dear Reviewer 5.

Thank You for the time and effort that you put into the improvement of the manuscript.

We hope that all corrections that we made will be satisfactory for You and that You will find the article suitable for the publication in the journal.  

  1. Line 10.... Please remove one of the two signs from :

Response: Done, thank you.

  1. Line 12….. BMI

Please spell out BMI

Response: Done, as recommended, thank you.

  1. Lines 16-17…. ERS/ATS recommendations

Please spell out ERS/ATS

Response: Done, as recommended, thank you.

  1. Line 17… Body mass or BMI?

Response: Body mass.

  1. Line 13…. The participants of the study were 506 school-aged children Line 16… FeNO was measured in 447 children according to ERS/ATS Why only in 447 if the total is 506 children? I needed to explain.

Response: For some children, it was too difficult to finish proper FeNO measurement, therefore we obtained correct measurements in 447 (out of 506) children. This had been indicated in the material and methods section, please kindly see lines 77-78.

  1. Line 18…. Results: In the study group there were 49.9% (n=223) boys and 50.1% (n=224) girls.

  506 school-aged children ……………223 +224 = 447

Response: Only the participants who had correct FeNO measurements were further analysed and the missing value was excluded from the analysis since the FeNO values were crucial for the  statistical analysis. However, we are very grateful for detailed review.

  1. There are 59 left to complete the total of 506... Why?

Response: As we mention earlier, we had to exclude those children who failed in FeNO measurements.

  1. Lines 20-21..... while the normal body mass was found in 71.3% (n=159) boys, overweight 11.6% (n=26) and obesity 20 17% (n=38) 159+26+38= 223…..boys And the girls?

Response: We corrected the sentence. Thank you.

  1. Lines 22-23-24……The FeNO values after adjusting for age, sex, BMI, and symptoms from respiratory system were depended on age and symptoms from respiratory system only. Please, in Materials and Methods, the age of the patients was not included....

Response: Thank you, we fixed that as suggested.

  1. Line 38…..between body mass and FeNO levels body mass or BMI ?

 Response: BMI, thank you.

  1. Line 39…well. [9-11]. Please remove the point after well

Response: Done, thank you.

  1. Lines 50-51…..The participants of the study were school-aged children who were …

 But, what is the school age?

 Response: We added the values to make it clear, thank you.

  1. Line 69…Body mass

 Body mass or BMI ?

 Response: Body mass, because it was measured by the device.

  1. Lines 77-78…In the study group (n=506) the technically acceptable (FeNO measurements) were 77 obtained in 447 children (89% of success rate). Please, how many boys and girls?

 Response: 34 boys and 25 girls. 506 – 447 = 59.

 Line 95....

  1. Please isn't it number 3?

Response: We checked the numbers of paragraphs, thank you.

Round 2

Reviewer 2 Report

Thanks to the authors for accepting the comments. I don't have any new.

Reviewer 3 Report

ok now for publication.